# DEEP NEURAL NETWORKS AS GAUSSIAN PROCESSES

**Jaehoon Lee**[*][†]**, Yasaman Bahri**[*][†]**, Roman Novak , Samuel S. Schoenholz,
Jeffrey Pennington , Jascha Sohl-Dickstein**

Google Brain
{jaehlee, yasamanb, romann, schsam, jpennin, jaschasd}@google.com

## ABSTRACT

It has long been known that a single-layer fully-connected neural network with an
i.i.d. prior over its parameters is equivalent to a Gaussian process (GP), in the limit
of infinite network width. This correspondence enables exact Bayesian inference
for infinite width neural networks on regression tasks by means of evaluating the
corresponding GP. Recently, kernel functions which mimic multi-layer random
neural networks have been developed, but only outside of a Bayesian framework.
As such, previous work has not identified that these kernels can be used as co-
variance functions for GPs and allow fully Bayesian prediction with a deep neural
network.

In this work, we derive the *exact equivalence* between infinitely wide *deep* net-
works and GPs. We further develop a computationally efficient pipeline to com-
pute the covariance function for these GPs. We then use the resulting GPs to per-
form Bayesian inference for wide deep neural networks on MNIST and CIFAR-
10. We observe that trained neural network accuracy approaches that of the corre-
sponding GP with increasing layer width, and that the GP uncertainty is strongly
correlated with trained network prediction error. We further find that test perfor-
mance increases as finite-width trained networks are made wider and more similar
to a GP, and thus that GP predictions typically outperform those of finite-width
networks. Finally we connect the performance of these GPs to the recent theory
of signal propagation in random neural networks.

## 1 INTRODUCTION

Deep neural networks have emerged in recent years as flexible parametric models which can fit
complex patterns in data. As a contrasting approach, Gaussian processes have long served as a
traditional nonparametric tool for modeling. An equivalence between these two approaches was
derived in Neal (1994a), for the case of one layer networks in the limit of infinite width. Neal
(1994a) further suggested that a similar correspondence might hold for deeper networks.

Consider a deep fully-connected neural network with i.i.d. random parameters. Each scalar output
of the network, an affine transformation of the final hidden layer, will be a sum of i.i.d. terms. As we
will discuss in detail below, in the limit of infinite width the Central Limit Theorem[1] implies that the
function computed by the neural network (NN) *is* a function drawn from a Gaussian process (GP).
In the case of single hidden-layer networks, the form of the kernel of this GP is well known (Neal
(1994a); Williams (1997)).

This correspondence implies that if we choose the hypothesis space to be the class of infinitely
wide neural networks, an i.i.d. prior over weights and biases can be replaced with a corresponding
GP prior over functions. As noted by (Williams, 1997), this substitution enables *exact* Bayesian
inference for regression using neural networks. The computation requires building the necessary
covariance matrices over the training and test sets and straightforward linear algebra computations.

---

[*]Both authors contributed equally to this work.

[†]Work done as a member of the Google AI Residency program (g.co/airesidency).

[1]Throughout this paper, we assume the conditions on the parameter distributions and nonlinearities are such
that the Central Limit Theorem will hold; for instance, that the weight variance is scaled inversely proportional
to the layer width.

In light of the resurgence in popularity of neural networks, it is timely to revisit this line of work. We delineate the correspondence between *deep* and wide neural networks and GPs and utilize it for Bayesian training of neural networks on regression tasks.

## 1.1 RELATED WORK

Our work touches on aspects of GPs, Bayesian learning, and compositional kernels. The correspondence between infinite neural networks and GPs was first noted by Neal (1994a;b). Williams (1997) computes analytic GP kernels for single hidden-layer neural networks with error function or Gaussian nonlinearities and noted the use of the GP prior for exact Bayesian inference in regression. Duvenaud et al. (2014) discusses several routes to building deep GPs and observes the degenerate form of kernels that are composed infinitely many times – a point we will return to Section 3.2 – but they do not derive the form of GP kernels as we do. Hazan & Jaakkola (2015) also discusses constructing kernels equivalent to infinitely wide deep neural networks, but their construction does not go beyond two hidden layers with nonlinearities.

Related work has also appeared outside of the GP context but in compositional kernel constructions. Cho & Saul (2009) derives compositional kernels for polynomial rectified nonlinearities, which includes the Sign and ReLU nonlinearities, and can be used in GPs; our manner of composing kernels matches theirs, though the context is different. Daniely et al. (2016) extends the construction of compositional kernels to neural networks whose underlying directed acyclic graph is of general form. They also prove, utilizing the formalism of dual activations, that compositional kernels originating from fully-connected topologies with the same nonlinearity become degenerate when composed infinitely many times. In a different context than compositional kernels, Poole et al. (2016); Schoenholz et al. (2017) study the same underlying recurrence relation for the specific case of fully-connected networks and bounded nonlinearities. They distinguish regions in hyperparameter space with different fixed points and convergence behavior in the recurrence relations. The focus in these works was to better understand the expressivity and trainability of deep networks.

Drawing inspiration from the multi-layer nature of deep neural networks, there is a line of work considering various approaches to stacking GPs, such as deep GPs (Lawrence & Moore (2007); Damianou & Lawrence (2013); Hensman & Lawrence (2014); Duvenaud et al. (2014); Bui et al. (2016)), which can give rise to a richer class of probabilistic models beyond GPs. This contrasts with our work, where we study GPs that are in direct correspondence with deep, infinitely wide neural networks. Krauth et al. (2016) has recently explored the performance of GP models with deep kernels given in Cho & Saul (2009), implemented with scalable approximations. However, they do not discuss the equivalence between deep neural networks and GPs with compositional kernels, which constitutes a conceptual contribution of our work. Furthermore, we note that the GP kernels in our work are more general than the compositional kernel construction outlined in Cho & Saul (2009) in two respects: (i) we are not limited to rectified polynomials but can deal with general nonlinearities, and (ii) we consider two additional hyperparameters in the kernels, which would correspond to the weight and bias parameter variances in a neural network. Finally, Gal & Ghahramani (2016) connects dropout in deep neural networks with approximate Bayesian inference in deep GPs.

Another series of recent works (Wilson et al. (2016b;a); Al-Shedivat et al. (2017)), termed *deep kernel learning*, utilize GPs with base kernels which take in features produced by a deep multilayer neural network, and train the resulting model end-to-end. Our work differs from these in that our GP corresponds to a multilayer neural network. Additionally, our GP kernels have many fewer parameters, and these parameters correspond to the hyperparameters of the equivalent neural network.

## 1.2 SUMMARY OF CONTRIBUTIONS

We begin by specifying the form of a GP which corresponds to a deep, infinitely wide neural network – hereafter referred to as the Neural Network GP (NNGP) – in terms of a recursive, deterministic computation of the kernel function. The prescription is valid for generic pointwise nonlinearities in fully-connected feedforward networks. We develop a computationally efficient method (Section 2.5) to compute the covariance function corresponding to deep neural networks with fixed hyperparameters.

In this work, as a first proof of concept of our NNGP construction, we focus on exact Bayesian inference for regression tasks, treating classification as regression on class labels. While less principled, least-squares classification performs well (Rifkin et al., 2003) and allows us to compare exact inference via a GP to prediction by a trained neural network on well-studied tasks (MNIST and CIFAR-10 classification). Note that it is possible to extend GPs to softmax classification with cross entropy loss (Williams & Barber (1998); Rasmussen & Williams (2006)), which we aim to investigate in future work.

We conduct experiments making Bayesian predictions on MNIST and CIFAR-10 (Section 3) and compare against NNs trained with standard gradient-based approaches. The experiments explore different hyperparameter settings of the Bayesian training including network depth, nonlinearity, training set size (up to and including the full dataset consisting of tens of thousands of images), and weight and bias variance. Our experiments reveal that the best NNGP performance is consistently competitive against that of NNs trained with gradient-based techniques, and the best NNGP setting, chosen across hyperparameters, often surpasses that of conventional training (Section 3, Table 1). We further observe that, with increasing network width, the performance of neural networks with gradient-based training approaches that of the NNGP computation, and that the GP uncertainty is strongly correlated with prediction error. Furthermore, the performance of the NNGP depends on the structure of the kernel, which can be connected to recent work on signal propagation in networks with random parameters (Schoenholz et al., 2017).

## 2 Deep, infinitely wide Neural Networks are drawn from GPs

We begin by specifying the correspondence between GPs and deep, infinitely wide neural networks, which hinges crucially on application of the Central Limit Theorem. We review the single-hidden layer case (Section 2.2) before moving to the multi-layer case (Section 2.3).

### 2.1 Notation

Consider an $L$-hidden-layer fully-connected neural network with hidden layers of width $N_l$ (for layer $l$) and pointwise nonlinearities $\phi$. Let $x \in \mathbb{R}^{d_{in}}$ denote the input to the network, and let $z^L \in \mathbb{R}^{d_{out}}$ denote its output. The $i$th component of the activations in the $l$th layer, post-nonlinearity and post-affine transformation, are denoted $x_i^l$ and $z_i^l$ respectively. We will refer to these as the post- and pre-activations. (We let $x_i^0 \equiv x_i$ for the input, dropping the Arabic numeral superscript, and instead use a Greek superscript $x^\alpha$ to denote a particular input $\alpha$). Weight and bias parameters for the $l$th layer have components $W_{ij}^l, b_i^l$, which are independent and randomly drawn, and we take them all to have zero mean and variances $\sigma_w^2/N_l$ and $\sigma_b^2$, respectively. $\mathcal{GP}(\mu, K)$ denotes a Gaussian process with mean and covariance functions $\mu(\cdot)$, $K(\cdot, \cdot)$, respectively.

### 2.2 Review of Gaussian Processes and Single-layer Neural Networks

We briefly review the correspondence between single-hidden layer neural networks and GPs (Neal (1994a;b); Williams (1997)). The $i$th component of the network output, $z_i^1$, is computed as,

$$z_i^1(x) = b_i^1 + \sum_{j=1}^{N_1} W_{ij}^1 x_j^1(x), \quad x_j^1(x) = \phi\left(b_j^0 + \sum_{k=1}^{d_{in}} W_{jk}^0 x_k\right), \tag{1}$$

where we have emphasized the dependence on input $x$. Because the weight and bias parameters are taken to be i.i.d., the post-activations $x_j^1, x_{j'}^1$ are independent for $j \neq j'$. Moreover, since $z_i^1(x)$ is a sum of i.i.d terms, it follows from the Central Limit Theorem that in the limit of infinite width $N_1 \to \infty$, $z_i^1(x)$ will be Gaussian distributed. Likewise, from the multidimensional Central Limit Theorem, any finite collection of $\{z_i^1(x^{\alpha=1}), ..., z_i^1(x^{\alpha=k})\}$ will have a joint multivariate Gaussian distribution, which is exactly the definition of a Gaussian process. Therefore we conclude that $z_i^1 \sim \mathcal{GP}(\mu^1, K^1)$, a GP with mean $\mu^1$ and covariance $K^1$, which are themselves independent of $i$. Because the parameters have zero mean, we have that $\mu^1(x) = \mathbb{E}\left[z_i^1(x)\right] = 0$ and,

$$K^1(x, x') \equiv \mathbb{E}\left[z_i^1(x) z_i^1(x')\right] = \sigma_b^2 + \sigma_w^2 \mathbb{E}\left[x_i^1(x) x_i^1(x')\right] \equiv \sigma_b^2 + \sigma_w^2 C(x, x'), \tag{2}$$

where we have introduced $C(x, x')$ as in Neal (1994a); it is obtained by integrating against the distribution of $W^0, b^0$. Note that, as any two $z_i^1, z_j^1$ for $i \neq j$ are joint Gaussian and have zero covariance, they are guaranteed to be independent despite utilizing the same features produced by the hidden layer.

## 2.3 GAUSSIAN PROCESSES AND DEEP NEURAL NETWORKS

The arguments of the previous section can be extended to deeper layers by induction. We proceed by taking the hidden layer widths to be infinite in succession ($N_1 \to \infty, N_2 \to \infty$, etc.) as we continue with the induction, to guarantee that the input to the layer under consideration is already governed by a GP. In Appendix C we provide an alternative derivation in terms of Bayesian marginalization over intermediate layers, which does not depend on the order of limits, in the case of a Gaussian prior on the weights. A concurrent work (de G. Matthews et al., 2018) further derives the convergence rate towards a GP if all layers are taken to infinite width simultaneously, but at different rates.

Suppose that $z_j^{l-1}$ is a GP, identical and independent for every $j$ (and hence $x_j^l(x)$ are independent and identically distributed). After $l-1$ steps, the network computes

$$z_i^l(x) = b_i^l + \sum_{j=1}^{N_l} W_{ij}^l x_j^l(x), \quad x_j^l(x) = \phi(z_j^{l-1}(x)). \tag{3}$$

As before, $z_i^l(x)$ is a sum of i.i.d. random terms so that, as $N_l \to \infty$, any finite collection $\{z_i^1(x^{\alpha=1}), ..., z_i^1(x^{\alpha=k})\}$ will have joint multivariate Gaussian distribution and $z_i^l \sim \mathcal{GP}(0, K^l)$. The covariance is

$$K^l(x, x') \equiv \mathbb{E}\left[z_i^l(x) z_i^l(x')\right] = \sigma_b^2 + \sigma_w^2 \, \mathbb{E}_{z_i^{l-1} \sim \mathcal{GP}(0, K^{l-1})}\left[\phi(z_i^{l-1}(x))\phi(z_i^{l-1}(x'))\right]. \tag{4}$$

By induction, the expectation in Equation 4 is over the GP governing $z_i^{l-1}$, but this is equivalent to integrating against the joint distribution of only $z_i^{l-1}(x)$ and $z_i^{l-1}(x')$. The latter is described by a zero mean, two-dimensional Gaussian whose covariance matrix has distinct entries $K^{l-1}(x, x')$, $K^{l-1}(x, x)$, and $K^{l-1}(x', x')$. As such, these are the only three quantities that appear in the result. We introduce the shorthand

$$K^l(x, x') = \sigma_b^2 + \sigma_w^2 \, F_\phi\left(K^{l-1}(x, x'), \, K^{l-1}(x, x), \, K^{l-1}(x', x')\right) \tag{5}$$

to emphasize the recursive relationship between $K^l$ and $K^{l-1}$ via a deterministic function $F$ whose form depends only on the nonlinearity $\phi$. This gives an iterative series of computations which can be performed to obtain $K^L$ for the GP describing the network's final output.

For the base case $K^0$, suppose $W_{ij}^0 \sim \mathcal{N}(0, \sigma_w^2/d_{\text{in}}), b_j^0 \sim \mathcal{N}(0, \sigma_b^2)$; we can utilize the recursion relating $K^1$ and $K^0$, where

$$K^0(x, x') = \mathbb{E}\left[z_j^0(x) z_j^0(x')\right] = \sigma_b^2 + \sigma_w^2\left(\frac{x \cdot x'}{d_{\text{in}}}\right). \tag{6}$$

In fact, these recurrence relations have appeared in other contexts. They are exactly the relations derived in the mean field theory of signal propagation in fully-connected random neural networks (Poole et al. (2016); Schoenholz et al. (2017)) and also appear in the literature on compositional kernels (Cho & Saul (2009); Daniely et al. (2016)). For certain activation functions, Equation 5 can be computed analytically (Cho & Saul (2009); Daniely et al. (2016)). In the case of the ReLU nonlinearity, it yields the well-known arccosine kernel (Cho & Saul (2009)) whose form we reproduce in Appendix B. When no analytic form exists, it can instead be efficiently computed numerically, as described in Section 2.5.

## 2.4 BAYESIAN TRAINING OF NEURAL NETWORKS USING GAUSSIAN PROCESS PRIORS

Here we provide a short review of how a GP prior over functions can be used to do Bayesian inference; see e.g. (Rasmussen & Williams, 2006) for a comprehensive review of GPs. Given a dataset

$\mathcal{D} = \{(x^1, t^1), ..., (x^n, t^n)\}$ consisting of input-target pairs $(x, t)$, we wish to make a Bayesian prediction at test point $x^*$ using a distribution over functions $z(x)$. This distribution is constrained to take values $\boldsymbol{z} \equiv (z^1, ..., z^n)$ on the training inputs $\boldsymbol{x} \equiv (x^1, ..., x^n)$ and,

$$P(z^*|\mathcal{D}, x^*) = \int d\boldsymbol{z}\, P(z^*|\boldsymbol{z}, \boldsymbol{x}, x^*)\, P(\boldsymbol{z}|\mathcal{D}) = \frac{1}{P(\mathbf{t})} \int d\boldsymbol{z}\, P(z^*, \boldsymbol{z}|x^*, \boldsymbol{x})\, P(\mathbf{t}|\boldsymbol{z}), \quad (7)$$

where $\boldsymbol{t} = (t^1, ..., t^n)^T$ are the targets on the training set, and $P(\mathbf{t}|\boldsymbol{z})$ corresponds to observation noise. We will assume a noise model consisting of a Gaussian with variance $\sigma_\epsilon^2$ centered at $\boldsymbol{z}$.

If the conditions of Section 2.2 or 2.3 apply, our choice of prior over functions implies that $z^1, ..., z^n, z^*$ are $n + 1$ draws from a GP and $z^*, \boldsymbol{z}|x^*, \boldsymbol{x} \sim \mathcal{N}(0, \mathbf{K})$ is a multivariate Gaussian whose covariance matrix has the form

$$\mathbf{K} = \begin{bmatrix} K_{\mathcal{D},\mathcal{D}} & K_{x^*,\mathcal{D}}^T \\ K_{x^*,\mathcal{D}} & K_{x^*,x^*} \end{bmatrix},$$

where the block structure corresponds to the division between the training set and the test point. That is, $K_{\mathcal{D},\mathcal{D}}$ is an $n \times n$ matrix whose $(i, j)$th element is $K(x^i, x^j)$ with $x^i, x^j \in \mathcal{D}$, while e.g. the $i$th element of $K_{x^*,\mathcal{D}}$ is $K(x^*, x^i), x^i \in \mathcal{D}$. As is standard in GPs, the integral in Equation 7 can be done exactly, resulting in $z^*|\mathcal{D}, x^* \sim \mathcal{N}(\bar{\mu}, \bar{K})$ with

$$\bar{\mu} = K_{x^*,\mathcal{D}}(K_{\mathcal{D},\mathcal{D}} + \sigma_\epsilon^2 \mathbb{I}_n)^{-1} \boldsymbol{t} \tag{8}$$

$$\bar{K} = K_{x^*,x^*} - K_{x^*,\mathcal{D}}(K_{\mathcal{D},\mathcal{D}} + \sigma_\epsilon^2 \mathbb{I}_n)^{-1} K_{x^*,\mathcal{D}}^T \tag{9}$$

where $\mathbb{I}_n$ is the $n \times n$ identity. The predicted distribution for $z^*|\mathcal{D}, x^*$ is hence determined from straightforward matrix computations, yet nonetheless corresponds to fully Bayesian training of the deep neural network. The form of the covariance function used is determined by the choice of GP prior, i.e. the neural network model class, which depends on depth, nonlinearity, and weight and bias variances. We henceforth resume placing a superscript $L$ as in $\mathbf{K}^L$ to emphasize the choice of depth for the compositional kernel.

## 2.5 Efficient Implementation of the GP Kernel

Given an $L$-layer deep neural network with fixed hyperparameters, constructing the covariance matrix $\mathbf{K}^L$ for the equivalent GP involves computing the Gaussian integral in Equation 4 for all pairs of training-training and training-test points, recursively for all layers. For some nonlinearities, such as ReLU, this integration can be done analytically. However, to compute the kernel corresponding to arbitrary nonlinearities, the integral must be performed numerically.

The most direct implementation of a numerical algorithm for $\mathbf{K}^L$ would be to compute integrals independently for each pair of datapoints and each layer. This is prohibitively expensive and costs $\mathcal{O}\left(n_g^2 L(n_{\text{train}}^2 + n_{\text{train}} n_{\text{test}})\right)$, where $n_g^2$ is the sampling density for the pair of Gaussian random variables in the 2D integral and $n_{\text{train}}, n_{\text{test}}$ are the training and test set sizes, respectively. However, by careful pipelining, and by preprocessing all inputs to have identical norm, we can improve this cost to $\mathcal{O}\left(n_g^2 n_v n_c + L(n_{\text{train}}^2 + n_{\text{train}} n_{\text{test}})\right)$, where $n_v$ and $n_c$ are sampling densities for a variance and correlation grid, as described below. In order to achieve this, we break the process into several steps:

1. Generate: pre-activations $u = [-u_{\max}, \cdots, u_{\max}]$ consisting of $n_g$ elements linearly spaced between $-u_{\max}$ and $u_{\max}$; variances $s = [0, \cdots, s_{\max}]$ with $n_v$ linearly spaced elements, where $s_{\max} < u_{\max}^2$; and correlations $c = (-1, \cdots, 1)$ with $n_c$ linearly spaced elements. Note that we are using fixed, rather than adaptive, sampling grids to allow operations to be parallelized and reused across datapoints and layers.

2. Populate a matrix $F$ containing a lookup table for the function $F_\phi$ in Equation 5. This involves numerically approximating a Gaussian integral, in terms of the marginal variances $s$ and correlations $c$. We guarantee that the marginal variance is identical for each datapoint, by preprocessing all datapoints to have identical norm at the input layer, so the number of

entries in the lookup table need only be $n_v n_c$. These entries are computed as[2]:

$$F_{ij} = \frac{\sum_{ab} \phi(u_a)\phi(u_b) \exp\left(-\frac{1}{2}\begin{bmatrix} u_a \\ u_b \end{bmatrix}^T \begin{bmatrix} s_i & s_i c_j \\ s_i c_j & s_i \end{bmatrix}^{-1} \begin{bmatrix} u_a \\ u_b \end{bmatrix}\right)}{\sum_{ab} \exp\left(-\frac{1}{2}\begin{bmatrix} u_a \\ u_b \end{bmatrix}^T \begin{bmatrix} s_i & s_i c_j \\ s_i c_j & s_i \end{bmatrix}^{-1} \begin{bmatrix} u_a \\ u_b \end{bmatrix}\right)}. \tag{10}$$

3. For every pair of datapoints $x$ and $x'$ in layer $l$, compute $K^l(x, x')$ using Equation 5. Approximate the function $F_\phi\left(K^{l-1}(x, x'); K^{l-1}(x, x); K^{l-1}(x', x')\right)$ by bilinear interpolation into the matrix $F$ from Step 2, where we interpolate into $s$ using the value of $K^{l-1}(x, x)$, and interpolate into $c$ using $\left(K^{l-1}(x, x')/K^{l-1}(x, x)\right)$. Remember that $K^{l-1}(x, x) = K^{l-1}(x', x')$, due to data preprocessing to guarantee constant norm.

4. Repeat the previous step recursively for all layers. Bilinear interpolation has constant cost, so this has cost $\mathcal{O}\left(L(n_{\text{train}}^2 + n_{\text{train}} n_{\text{test}})\right)$.

This computational recipe allows us to compute the covariance matrix for the NNGP corresponding to any well-behaved nonlinearity $\phi$. All computational steps above can be implemented using accelerated tensor operations, and computation of $\mathbf{K}^L$ is typically faster than solving the system of linear equations in Equation 8-9. Figure 6 illustrates the close agreement between the kernel function computed numerically (using this approach) and analytically, for the ReLU nonlinearity. It also illustrates the angular dependence of the kernel and its evolution with increasing depth.

Finally, note that the full computational pipeline is *deterministic* and *differentiable*. The shape and properties of a deep network kernel are purely determined by hyperparameters of the deep neural network. Since GPs give exact marginal likelihood estimates, this kernel construction may allow principled hyperparameter selection, or nonlinearity design, e.g. by gradient ascent on the log likelihood w.r.t. the hyperparameters. Although this is not the focus of current work, we hope to return to this topic in follow-up work.

An open source implementation of the algorithm is available at https://github.com/brain-research/nngp.

# 3 EXPERIMENTAL RESULTS

## 3.1 DESCRIPTION

We compare NNGPs with SGD[3] trained neural networks on the permutation invariant MNIST and CIFAR-10 datasets. The baseline neural network is a fully-connected network with identical width at each hidden layer. Training is on the mean squared error (MSE) loss, chosen so as to allow direct comparison to GP predictions. Formulating classification as regression often leads to good results (Rifkin & Klautau, 2004). Future work may involve evaluating the NNGP on a cross entropy loss using the approach in (Williams & Barber, 1998; Rasmussen & Williams, 2006). Training used the Adam optimizer (Kingma & Ba (2014)) with learning rate and initial weight/bias variances optimized over validation error using the Google Vizier hyperparameter tuner (Golovin et al., 2017). Dropout was not used. In future work, it would be interesting to incorporate dropout into the NNGP covariance matrix using an approach like that in (Schoenholz et al., 2017). For the study, nonlinearities were chosen to be either rectified linear units (ReLU) or hyperbolic tangent (Tanh). Class labels were encoded as a one-hot, zero-mean, regression target (i.e., entries of -0.1 for the incorrect class and 0.9 for the correct class). We constructed the covariance kernel numerically for ReLU and Tanh nonlinearities following the method described in Section 2.5.

**Performance**: We find that the NNGP often outperforms trained finite width networks. See Table 1 and Figure 1.

---

[2] For numerical reasons, in practice an independent 1D lookup table is built for the case that $c_j = 1$.

[3] For all presented results, the variant of SGD used is Adam. Although not shown, we found vanilla SGD produced qualitatively similar results, with slightly higher MSE.

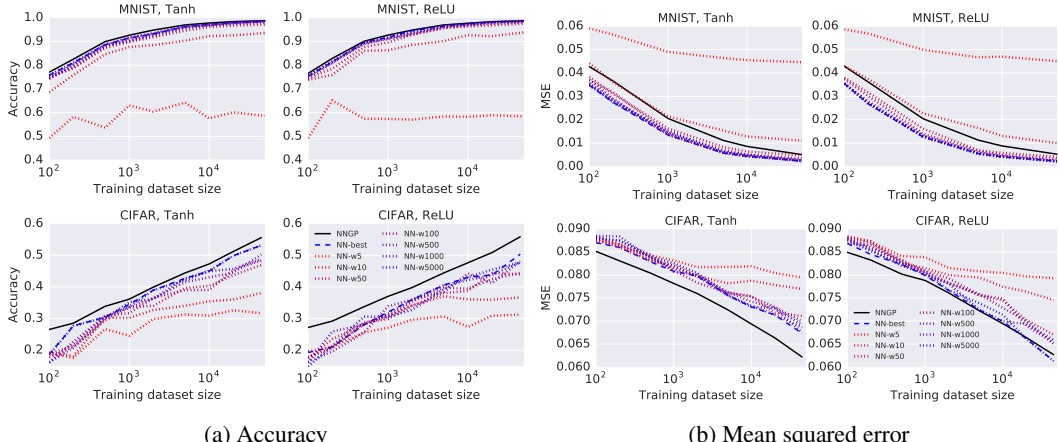

(a) Accuracy            (b) Mean squared error

Figure 1: The NNGP often outperforms finite width networks, and neural network performance more closely resembles NNGP performance with increasing width. Test accuracy and mean squared error on MNIST and CIFAR-10 dataset are shown for the best performing NNGP and best performing SGD trained neural networks for given width. 'NN-best' denotes the best performing (on the validation set) neural network across all widths and trials. Often this is the neural network with the largest width.

We additionally find the performance of the best finite-width NNs, trained with a variant of SGD, approaches that of the NNGP with increasing layer width. This is interesting from at least two, potentially related, standpoints. (1) NNs are commonly believed to be powerful because of their ability to do flexible representation learning, while our NNGP uses fixed basis functions; nonetheless, in our experiments we find no salient performance advantage to the former. (2) It hints at a possible relationship between SGD and Bayesian inference in certain regimes – were the neural networks trained in a fully Bayesian fashion, rather than by SGD, the approach to NNGP in the large width limit would be guaranteed. There is recent work suggesting that SGD can implement approximate Bayesian inference (Mandt et al., 2017) under certain assumptions.

The similarity of the performance of the widest NN in Figure 1 with the NNGP suggests that the limit of infinite network width, which is inherent to the GP, is far from being a disadvantage. Indeed, in practice it is found that the best generalizing NNs are in fact the widest. To support this, in Figure 2 we show generalization gap results from an experiment in which we train 180 fully-connected networks with five hidden layers on CIFAR-10 with a range of layer widths. For this experiment, we trained the networks using a standard cross entropy loss rather than MSE, leading to a slight difference in performance.

**Uncertainty**: One benefit in using a GP is that, due to its Bayesian nature, all predictions have uncertainty estimates (Equation 9). For conventional neural networks, capturing the uncertainty in a model's predictions is challenging (Gal, 2016). In the NNGP, every test point has an explicit estimate of prediction variance associated with it (Equation 9). In our experiments, we observe that the NNGP uncertainty estimate is highly correlated with prediction error (Figure 3).

### 3.2 RELATIONSHIP TO DEEP SIGNAL PROPAGATION

Several prior works (Poole et al. (2016); Schoenholz et al. (2017); Daniely et al. (2016); Duvenaud et al. (2014)) have noted the recurrence relations Equation 5 commonly approach a functionally uninteresting fixed point with depth $l \to \infty$, in that $K^\infty(x, x')$ becomes a constant or piecewise constant map. We now briefly relate our ability to train NNGPs with the convergence of $K^l(x, x')$ to the fixed-point kernel.

We will be particularly interested in contextualizing our results in relation to Poole et al. (2016); Schoenholz et al. (2017) which analyzed the fixed points and the approach to them in detail for bounded nonlinearities. To briefly recapitulate: there are regions of hyperparameter space (called phases) where $K^\infty(x, x')$ changes only quantitatively with $\sigma_w^2$ and $\sigma_b^2$. However, there are low

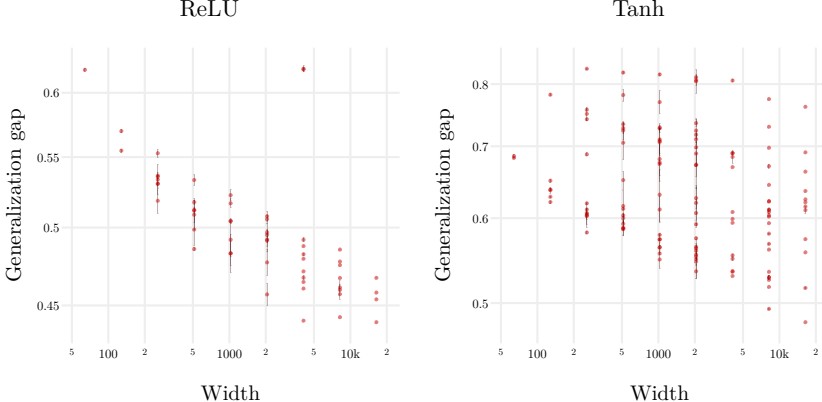

Figure 2: Generalization gap for five hidden layer fully-connected networks with variable widths, using ReLU and Tanh nonlinearities on CIFAR-10. Random optimization and initialization hyperparameters were used and results were filtered for networks with 100% classification training accuracy, resulting in a total of 125 Tanh and 55 ReLU networks. The best generalizing networks are consistently the widest.

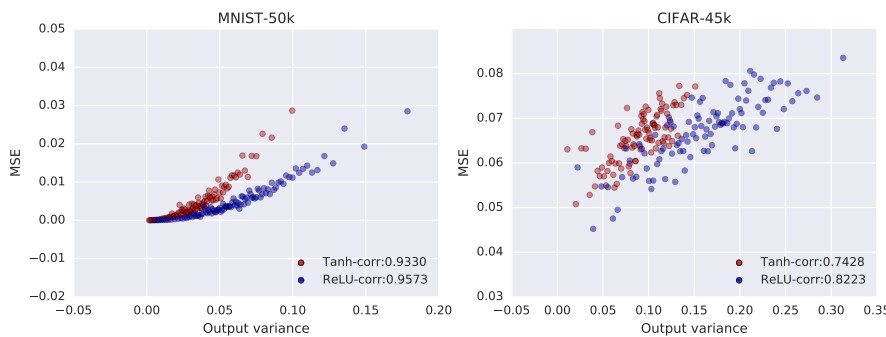

Figure 3: The Bayesian nature of NNGP allows it to assign a prediction uncertainty to each test point. This prediction uncertainty is highly correlated with the empirical error on test points. The $x-$axis shows the predicted MSE for test points, while the $y-$axis shows the realized MSE. To allow comparison of *mean* squared error, each plotted point is an average over 100 test points, binned by predicted MSE. The hyperparameters for the NNGP are depth$= 3$, $\sigma_w^2 = 2.0$, and $\sigma_b^2 = 0.2$. See Appendix Figure 8 for dependence on training set size.

dimensional boundaries that separate different phases and between them the nature of $K^\infty(x, x')$ changes qualitatively.

For the Tanh nonlinearity, there are two distinct phases respectively called the "ordered" phase and the "chaotic" phase that can be understood as a competition between the weights and the biases of the network. A diagram showing these phases and the boundary between them is shown in Figure 4a. In the ordered phase, the features obtained by propagating an input through the each layer of the recursion become similar for dissimilar inputs. Fundamentally, this occurs because the different inputs share common bias vectors and so all inputs end up just approaching the random bias. In this case the covariance $K^l(x, x') \to q^*$ for every pair of inputs $x, x'$, where $q^*$ is a constant that depends only on $\sigma_w^2$ and $\sigma_b^2$. All inputs have unit correlation asymptotically with depth. By contrast in the chaotic phase the weight variance $\sigma_w^2$ dominates and similar inputs become dissimilar with depth as they are randomly projected by the weight matrices. In this case, the covariance $K^l(x, x') \to q^*$ for $x = x'$ but $q^* c^*$ for $x \neq x'$. Here $c^* < 1$ is the fixed point correlation. In each of these regimes, there is also a finite depth-scale $\xi$ which describes the characteristic number of layers over which the covariance function decays exponentially towards its fixed point form. Exactly at the boundary between these two regimes is a line in $(\sigma_w^2, \sigma_b^2)$-space where the decay $K^l(x, x')$ towards its fixed

Table 1: The NNGP often outperforms finite width networks. Test accuracy on MNIST and CIFAR-10 datasets. The reported NNGP results correspond to the best performing depth, $\sigma_w^2$, and $\sigma_b^2$ values on the validation set. The traditional NN results correspond to the best performing depth, width and optimization hyperparameters. Best models for a given training set size are specified by (depth-width-$\sigma_w^2$-$\sigma_b^2$) for NNs and (depth–$\sigma_w^2$-$\sigma_b^2$) for GPs. More results are in Appendix Table 2.

| Num training | Model (ReLU) | Test accuracy | Model (tanh) | Test accuracy |
|---|---|---|---|---|
| MNIST:1k | NN-2-5000-3.19-0.00 | 0.9252 | NN-2-1000-0.60-0.00 | 0.9254 |
| | GP-20-1.45-0.28 | **0.9279** | GP-20-1.96-0.62 | 0.9266 |
| MNIST:10k | NN-2-2000-0.42-0.16 | 0.9771 | NN-2-2000-2.41-1.84 | 0.9745 |
| | GP-7-0.61-0.07 | 0.9765 | GP-2-1.62-0.28 | **0.9773** |
| MNIST:50k | NN-2-2000-0.60-0.44 | 0.9864 | NN-2-5000-0.28-0.34 | 0.9857 |
| | GP-1-0.10-0.48 | 0.9875 | GP-1-1.28-0.00 | **0.9879** |
| CIFAR:1k | NN-5-500-1.29-0.28 | 0.3225 | NN-1-200-1.45-0.12 | 0.3378 |
| | GP-7-1.28-0.00 | 0.3608 | GP-50-2.97-0.97 | **0.3702** |
| CIFAR:10k | NN-5-2000-1.60-1.07 | 0.4545 | NN-1-500-1.48-1.59 | 0.4429 |
| | GP-5-2.97-0.28 | **0.4780** | GP-7-3.48-2.00 | 0.4766 |
| CIFAR:45k | NN-3-5000-0.53-0.01 | 0.5313 | NN-2-2000-1.05-2.08 | 0.5034 |
| | GP-3-3.31-1.86 | **0.5566** | GP-3-3.48-1.52 | 0.5558 |

point is significantly slower and non-exponential. It was noted in Schoenholz et al. (2017) that this approach to the fixed-point covariance fundamentally bounded whether or not neural networks could successfully be trained. It was shown that initializing networks on this line allowed for significantly deeper neural networks to be trained.

For ReLU networks a similar picture emerges, however there are some subtleties due to the unbounded nature of the nonlinearity. In this case for all $\sigma_w^2$ and $\sigma_b^2$, $K^\infty(x, x') = q^*$ for all $x, x'$ and every point becomes asymptotically correlated. Despite this, there are again two phases: a "bounded" phase in which $q^*$ is finite (and nonzero) and an unbounded phase in which $q^*$ is either infinite or zero. As in the Tanh case there are depth scales that control the rate of convergence to these fixed points and therefore limit the maximum trainable depth. The phase diagram for the ReLU nonlinearity is also shown in Figure 4b.

In a striking analogy with the trainability of neural networks, we observe that the performance of the NNGP appears to closely track the structure from the phase diagram, clearly illustrated in Figure 4. Indeed, we see that as for hyperparameter settings that are far from criticality, the GP is unable to train and we encounter poor test set performance. By contrast, near criticality we observe that our models display high accuracy. Moreover, we find that the accuracy appears to drop more quickly away from the phase boundary with increase in depth $L$ of the GP kernel, $K^L$. To understand this effect we note that information about data will be available to our model only through the difference $K^L(x, x') - K^\infty(x, x')$. However, as the depth gets larger, this difference becomes increasingly small and at some point can no longer be represented due to numerical precision. At this point our test accuracy begins to quickly degrade to random chance.

## 4 CONCLUSION AND FUTURE DIRECTIONS

By harnessing the limit of infinite width, we have specified a correspondence between priors on deep neural networks and Gaussian processes whose kernel function is constructed in a compositional, but fully deterministic and differentiable, manner. Use of a GP prior on functions enables exact Bayesian inference for regression from matrix computations, and hence we are able to obtain predictions and uncertainty estimates from deep neural networks without stochastic gradient-based training. The performance is competitive with the best neural networks (within specified class of fully-connected models) trained on the same regression task under similar hyperparameter settings. While we were able to run experiments for somewhat large datasets (sizes of 50k), we intend to look into scalability for larger learning tasks, possibly harnessing recent progress in scalable GPs (Quiñonero-Candela & Rasmussen (2005); Hensman et al. (2013)).

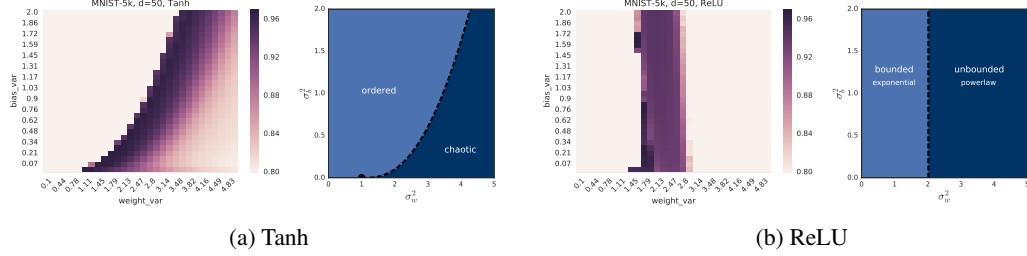

(a) Tanh                                         (b) ReLU

Figure 4: The best performing NNGP hyperparameters agree with those predicted by deep signal propagation. Test set accuracy heatmaps for NNGPs evaluated for a grid of $\sigma_w^2$ and $\sigma_b^2$ values. The right plot in each subfigure (a), (b) is a theoretical phase diagram for that nonlinearity following the methodology of Schoenholz et al. (2017). We observe that the performance of the NNGP is best along the critical line (dotted lines). Additional depths are shown in the Appendix Figure 9.

In our experiments, we observed the performance of the optimized neural network appears to approach that of the GP computation with increasing width. Whether gradient-based stochastic optimization implements an approximate Bayesian computation is an interesting question (Mandt et al., 2017). Further investigation is needed to determine if SGD does approximately implement Bayesian inference under the conditions typically employed in practice.

Additionally, the NNGP provides explicit estimates of uncertainty. This may be useful in predicting model failure in critical applications of deep learning, or for active learning tasks where it can be used to identify the best datapoints to hand label.

ACKNOWLEDGMENTS

We thank Ryan Adams, Samy Bengio, and Matt Hoffman for useful discussions and feedback, and Gamaleldin Elsayed and Daniel Levy for helpful comments on the manuscript.

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

## A DRAWS FROM AN NNGP PRIOR

Figure 5 illustrates the nature of the GP prior for the ReLU nonlinearity by depicting samples of 1D functions $z(x)$ drawn from a ReLU GP, $\mathcal{GP}(0, K^L)$, with fixed depth $L = 10$ and $(\sigma_w^2, \sigma_b^2) = (1.8, 0.01)$.

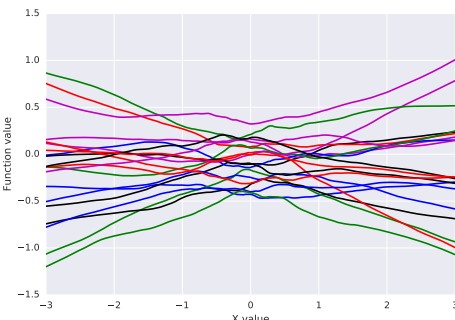

Figure 5: Samples from an NNGP prior for 1D functions. Different lines correspond to different draws (arbitrary colors).

## B ANALYTIC FORM FOR KERNEL AND COMPARISON

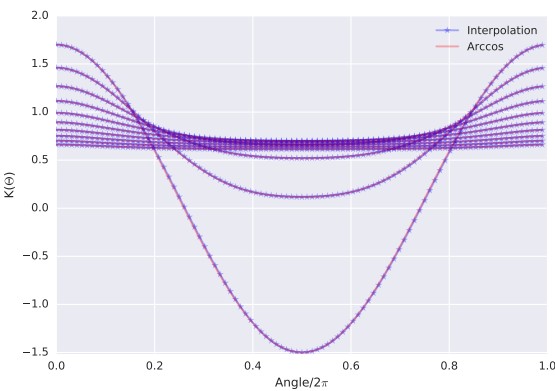

Figure 6: The angular structure of the kernel and its evolution with depth. Also illustrated is the good agreement between the kernel computed using the methods of Section 2.5 (blue, starred) and the analytic form of the kernel (red). The depth $l$ in $K^l$ runs from $l = 0, ..., 9$ (flattened curves for increasing $l$), and $(\sigma_w^2, \sigma_b^2) = (1.6, 0.1)$.

In the main text, we noted that the recurrence relation Equation 5 can be computed analytically for certain nonlinearities. In particular, this was computed in Cho & Saul (2009) for polynomial rectified nonlinearities. For ReLU, the result including the weight and bias variance is

$$K^l(x, x') = \sigma_b^2 + \frac{\sigma_w^2}{2\pi}\sqrt{K^{l-1}(x,x)K^{l-1}(x',x')}\left(\sin\theta_{x,x'}^{l-1} + (\pi - \theta_{x,x'}^{l-1})\cos\theta_{x,x'}^{l-1}\right),$$

$$\theta_{x,x'}^l = \cos^{-1}\left(\frac{K^l(x,x')}{\sqrt{K^l(x,x)K^l(x',x')}}\right). \tag{11}$$

To illustrate the angular form of $K^l(x, x')$ and its evolution with $l$, in Figure 6 we plot $K^l(\theta)$ for the ReLU nonlinearity, where $\theta$ is the angle between $x$ and $x'$ with norms such that $||x||^2 = ||x'||^2 =$

$d_{\text{in}}$. We observe a flattening of the angular structure with increase in depth $l$, as predicted from the understanding in Section 3.2. Simultaneously, the figure also illustrates the good agreement between the kernel computed using the numerical implementation of Section 2.5 (blue, starred) and the analytic arccosine kernel, Equation 11 (red), for a particular choice of hyperparameters $(\sigma_w^2, \sigma_b^2)$.

## C   BAYESIAN MARGINALIZATION OVER INTERMEDIATE LAYERS

In this section, we present an alternate derivation of the equivalence between infinitely wide deep neural networks and Gaussian process by marginalization over intermediate layers. For this derivation, we take the weight and bias parameters to be drawn from independent Gaussians, with zero mean and appropriately scaled variance.

We are interested in finding the distribution $p(z^L|x)$ over network outputs $z^L \in \mathcal{R}^{d_{\text{out}} \times B}$, conditioned on network inputs $x \in \mathcal{R}^{d_{\text{in}} \times B}$, for input dimensionality $d_{\text{in}}$, output dimensionality $d_{\text{out}}$, and dataset size $B$. Intervening layers will have width $N_l$, $z^l \in \mathcal{R}^{N_{l+1} \times B}$ for $L > l > 0$. We define the second moment matrix (here *post*-nonlinearity) for each layer $l$ to be

$$K_{ab}^l = \begin{cases} \frac{1}{d_{\text{in}}} \sum_n x_{na} x_{nb} & l = 0 \\ \frac{1}{N_l} \sum_n \phi(z_{na}^{l-1}) \phi(z_{nb}^{l-1}) & l > 0 \end{cases}. \tag{12}$$

Our approach is to think of intermediate random variables corresponding to these second moments defined above. By definition, $K^l$ only depends on $z^{l-1}$. In turn, the pre-activations $z^l$ are described by a Gaussian process conditioned on the second moment matrix $K^l$,

$$\begin{aligned} p(z^l|K^l) &= \mathcal{N}\left(\text{vec}\left(z^l\right); 0, G\left(K^l\right) \otimes I\right) \\ &=: \mathcal{GP}\left(z^l; 0, G\left(K^l\right)\right), \end{aligned} \tag{13}$$

where

$$G\left(K^l\right) := \sigma_w^2 K^l + \sigma_b^2 \mathbf{1}\mathbf{1}^T. \tag{14}$$

This correspondence of each layer to a GP, conditioned on the layer's second moment matrix, is exact even for finite width $N_l$ because the parameters are drawn from a Gaussian. Altogether, this justifies the graphical model depicted in Figure 7.

We will write $p(z^L|x)$ as an integral over all the intervening second moment matrices $K^l$,

$$p(z^L|x) = \int p\left(z^L, K^0, K^1, \cdots, K^L|x\right) dK^{0\cdots L}. \tag{15}$$

This joint distribution can be decomposed as

$$p(z^L|x) = \int p(z^L|K^L) \left(\prod_{l=1}^L p(K^l|K^{l-1})\right) p(K^0|x) dK^{0\cdots L}. \tag{16}$$

The directed decomposition in Equation 16 holds because $K^L$ is a function only of $z^{L-1}$ (see Equation 12), $z^{L-1}$ depends only on $K^{L-1}$ (see Equation 13), $K^{L-1}$ is a function only of $z^{L-2}$, etc (Figure 7).

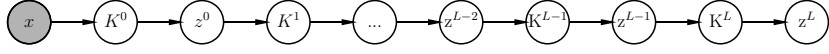

Figure 7: Graphical model for neural network's computation.

The sum in Equation 12 for $l > 0$ is a sum over i.i.d. terms. As $N_l$ grows large, the Central Limit Theorem applies, and $p\left(K^l|K^{l-1}\right)$ converges to a Gaussian with variance that shrinks as $\frac{1}{N_l}$. Further, in the infinite width limit it will go to a delta function,

$$\lim_{N_l \to \infty} p\left(K^l|K^{l-1}\right) = \delta\left(K^l - (F \circ G)\left(K^{l-1}\right)\right), \tag{17}$$

with $F(\cdot)$ defined as in Equation 5. Similarly, the dependence of $K^0$ on $x$ can be expressed as a delta function,

$$p\left(K^0|x\right) = \delta\left(K^0 - \frac{1}{d_{\text{in}}}x^T x\right). \tag{18}$$

Substituting $p(z^L|K^L)$, $p\left(K^l|K^{l-1}\right)$ and $p\left(K^0|x\right)$ into Equation 16, we get

$$\lim_{N_L \to \infty, \ldots, N_1 \to \infty} p(z^L|x) = \int \mathcal{GP}\left(z^L; 0, G\left(K^L\right)\right) \left(\prod_{l=1}^{L} \delta\left(K^l - (F \circ G)\left(K^{l-1}\right)\right)\right) \delta\left(K^0 - \frac{1}{d_{\text{in}}}x^T x\right) dK^{0\cdots L}$$

$$= \mathcal{GP}\left(z^L; 0, \left(G \circ (F \circ G)^L\right)\left(\frac{1}{d_{\text{in}}}x^T x\right)\right)$$

$$= \mathcal{GP}\left(z^L; 0, \left(G \circ (F \circ G)^L\right)\left(K^0\right)\right). \tag{19}$$

So, in the limit of infinite width, $z^L|x$ is described by a Gaussian process with kernel $\left(G \circ (F \circ G)^L\right)\left(K^0\right)$.

## D    DETAILS OF THE EXPERIMENTS

We outline details of the experiments for Section 3. For MNIST we use a 50k/10k/10k split of the training/validation/test dataset. For CIFAR-10, we used a 45k/5k/10k split. The validation set was used for choosing the best hyperparameters and evaluation on the test set is reported.

For training neural networks hyperparameters were optimized via random search on average 250 trials for each choice of ($n_{\text{train}}$, depth, width, nonlinearity).

**Random search range**: Learning rate was sampled within $(10^{-4}, 0.2)$ in log-scale, weight decay constant was sampled from $(10^{-8}, 1.0)$ in log-scale, $\sigma_w \in [0.01, 2.5]$, $\sigma_b \in [0, 1.5]$ was uniformly sampled and mini-batch size was chosen equally among $[16, 32, 64, 128, 256]$.

For the GP with given depth and nonlinearity, a grid of 30 points evenly spaced from 0.1 to 5.0 (for $\sigma_w^2$) and 30 points evenly spaced from 0 to 2.0 (for $\sigma_b^2$) was evaluated to generate the heatmap. The best GP run was chosen among the 900 evaluations in the $\sigma_w^2$-$\sigma_b^2$ grid.

**Computation time**: We report computation times for NNGP experiments. The grid generation with took 440-460s with 6 CPUs for $n_g = 501, n_v = 501, n_c = 500$, which was amortized over all the experiments. For full (50k) MNIST, constructing $K_{DD}$ for each layer took 90-140s (depending on CPU generation) running on 64 CPUs. Solving linear equations via Cholesky decomposition took 180-220s for 1000 test points.

**Details of NNGP implementaion**: For all the experiments we used pre-computed lookup tables $F$ with $n_g = 501, n_v = 501, n_c = 500$, and $s_{max} = 100$. Default value for the target noise $\sigma_\epsilon^2$ was set to $10^{-10}$ and was increased by factor of 10 when Cholesky decomposition failed while solving Equation 8 and 9. We refer to Rasmussen & Williams (2006) for standard numerically stable implementation of GP regression.

## E    FURTHER RESULTS

Here we include more results from experiments described in Section 3.

**Uncertainty**: Relationship between the target MSE and the GP's uncertainty estimate for smaller training set size is shown in Figure 8.

**Performance**: Performance of grid points of $\sigma_w^2$-$\sigma_b^2$ for varying depth is shown in Figure 9. The best performing NNGP's hyperparameters are distributed near the critical line (Figure 10) where the phase changes as described in Section 3.2.

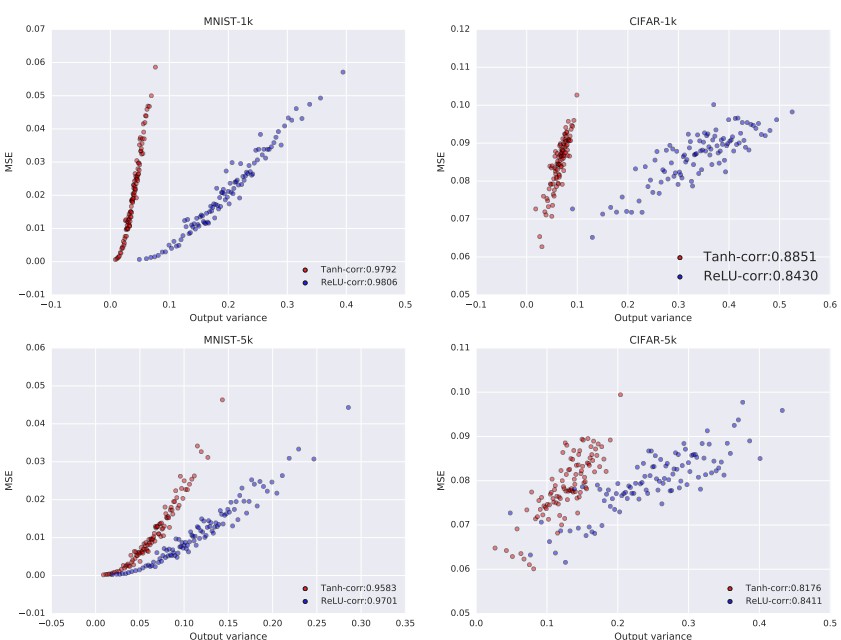

Figure 8: The prediction uncertainty for smaller number of training points. The details are the same as Figure 3.

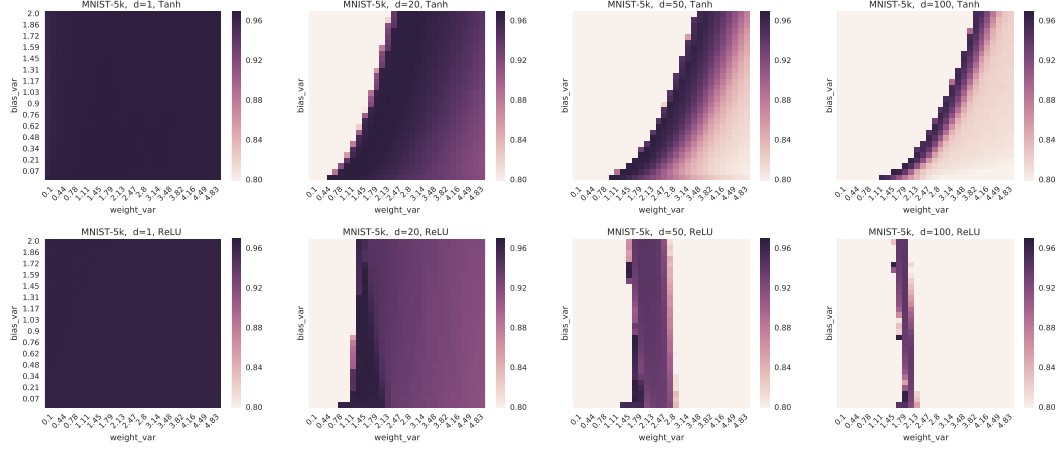

Figure 9: Test set accuracy heatmaps for NNGPs evaluated for a grid of $\sigma_w^2$ and $\sigma_b^2$ values for varying depth. Rows correspond to Tanh and ReLU nonlinearities, and columns correspond to varying depth.

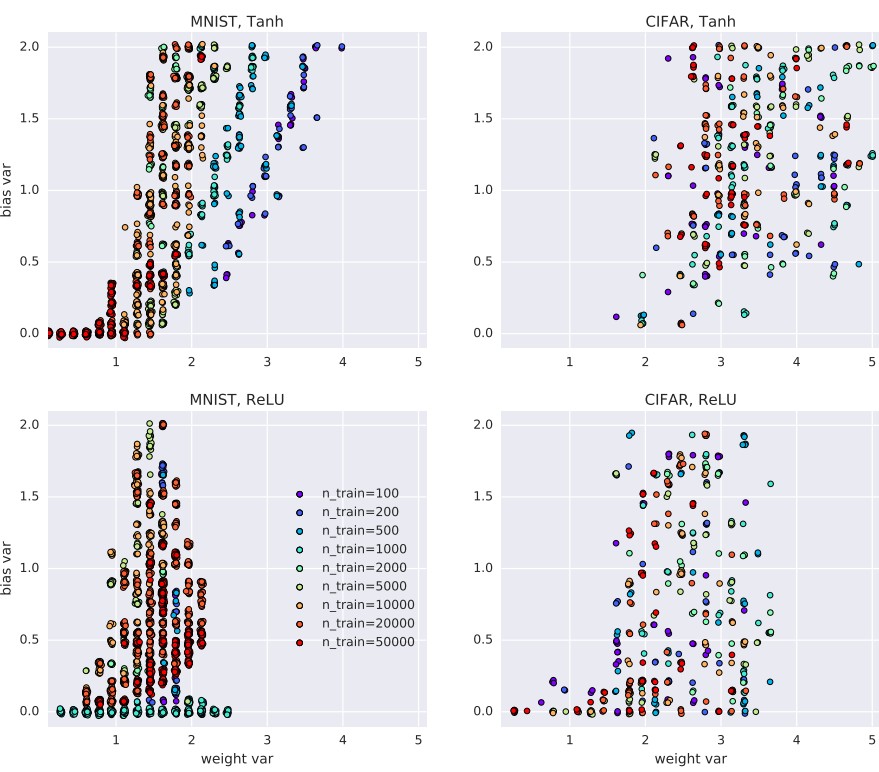

Figure 10: Best performing NNGPs are distributed near the critical line. Weight and bias variance distribution for the 25 best performing runs for NNGP with the given training set size is shown.

Table 2: Completion of Table 1. The reported NNGP results correspond to the best performing depth, $\sigma_w^2$, and $\sigma_b^2$ values on the validation set. The traditional NN results correspond to the best performing depth, width and optimization hyperparameters. Best models for a given training set size are specified by (depth-width-$\sigma_w^2$-$\sigma_b^2$) for NNs and (depth–$\sigma_w^2$-$\sigma_b^2$) for GPs.

| Num training | Model (ReLU) | Test accuracy | Model (tanh) | Test accuracy |
|---|---|---|---|---|
| MNIST:100 | NN-2-5000-0.10-0.11 | **0.7786** | NN-1-500-1.48-0.61 | 0.7766 |
| | GP-100-1.79-0.83 | 0.7735 | GP-100-3.14-0.97 | 0.7736 |
| MNIST:200 | NN-2-2000-0.52-0.00 | **0.8298** | NN-2-1000-1.80-1.99 | 0.8223 |
| | GP-100-1.79-0.83 | 0.8282 | GP-100-3.99-2.00 | 0.8277 |
| MNIST:500 | NN-2-5000-1.82-0.77 | **0.9028** | NN-1-5000-3.74-2.18 | 0.9001 |
| | GP-100-1.79-0.83 | 0.8995 | GP-50-3.48-1.86 | 0.9008 |
| MNIST:1k | NN-2-5000-3.19-0.00 | 0.9252 | NN-2-1000-0.60-0.00 | 0.9254 |
| | GP-20-1.45-0.28 | **0.9279** | GP-20-1.96-0.62 | 0.9266 |
| MNIST:2k | NN-2-5000-2.88-0.01 | 0.9468 | NN-1-2000-0.98-1.30 | 0.9462 |
| | GP-10-1.11-0.55 | **0.9485** | GP-10-1.79-1.45 | 0.9477 |
| MNIST:5k | NN-3-500-2.92-0.22 | 0.9675 | NN-2-1000-4.12-2.18 | 0.9655 |
| | GP-7-0.61-0.07 | 0.9692 | GP-3-1.11-0.00 | **0.9693** |
| MNIST:10k | NN-2-2000-0.42-0.16 | 0.9771 | NN-2-2000-2.41-1.84 | 0.9745 |
| | GP-7-0.61-0.07 | 0.9765 | GP-2-1.62-0.28 | **0.9773** |
| MNIST:20k | NN-3-1000-2.45-0.98 | 0.9825 | NN-2-2000-0.21-0.10 | 0.9814 |
| | GP-5-1.62-0.83 | 0.9830 | GP-1-2.63-0.00 | **0.9836** |
| MNIST:50k | NN-2-2000-0.60-0.44 | 0.9864 | NN-2-5000-0.28-0.34 | 0.9857 |
| | GP-1-0.10-0.48 | 0.9875 | GP-1-1.28-0.00 | **0.9879** |
| CIFAR:100 | NN-5-500-1.88-1.00 | 0.2586 | NN-2-200-3.22-2.09 | 0.2470 |
| | GP-3-4.49-0.97 | 0.2673 | GP-10-3.65-1.17 | **0.2718** |
| CIFAR:200 | NN-3-200-0.17-0.00 | 0.2719 | NN-3-200-1.41-0.21 | 0.2686 |
| | GP-3-3.99-1.72 | **0.3022** | GP-7-3.65-0.55 | 0.2927 |
| CIFAR:500 | NN-1-100-1.26-0.63 | 0.3132 | NN-1-2000-0.11-0.90 | 0.2939 |
| | GP-20-1.79-0.21 | **0.3395** | GP-7-3.65-0.62 | 0.3291 |
| CIFAR:1k | NN-5-500-1.29-0.28 | 0.3225 | NN-1-200-1.45-0.12 | 0.3378 |
| | GP-7-1.28-0.00 | 0.3608 | GP-50-2.97-0.97 | **0.3702** |
| CIFAR:2k | NN-3-5000-5.59-0.57 | 0.3894 | NN-5-1000-0.86-1.28 | 0.3597 |
| | GP-3-4.16-1.17 | 0.3953 | GP-5-4.66-1.03 | **0.3959** |
| CIFAR:5k | NN-5-2000-5.26-1.74 | 0.4241 | NN-1-5000-0.07-0.22 | 0.3993 |
| | GP-3-4.66-1.03 | **0.4454** | GP-10-3.65-1.38 | 0.4430 |
| CIFAR:10k | NN-5-2000-1.60-1.07 | 0.4545 | NN-1-500-1.48-1.59 | 0.4429 |
| | GP-5-2.97-0.28 | **0.4780** | GP-7-3.48-2.00 | 0.4766 |
| CIFAR:20k | NN-3-5000-4.18-0.18 | 0.5041 | NN-2-5000-0.02-1.12 | 0.4565 |
| | GP-3-5.00-0.83 | 0.5118 | GP-7-3.14-1.93 | **0.5124** |
| CIFAR:45k | NN-3-5000-0.53-0.01 | 0.5313 | NN-2-2000-1.05-2.08 | 0.5034 |
| | GP-3-3.31-1.86 | **0.5566** | GP-3-3.48-1.52 | 0.5558 |

