# OpenReview forum: "Deep Neural Networks as Gaussian Processes"
_ICLR.cc/2018/Conference — Accept (Poster)_

### Official Review · AnonReviewer1 · 2017-11-27
**Interesting paper but preliminary**

**Rating:** 4
**Confidence:** 4

**Review:**

Neal (1994) showed that a one hidden layer Bayesian neural network, under certain conditions, converges to a Gaussian process as the number of hidden units approaches infinity. Neal (1994) and Williams (1997) derive the resulting kernel functions for such Gaussian processes when the neural networks have certain transfer functions.

Similarly, the authors show an analogous result for deep neural networks with multiple hidden layers and an infinite number of hidden units per layer, and show the form of the resulting kernel functions. For certain transfer functions, the authors perform a numerical integration to compute the resulting kernels. They perform experiments on MNIST and CIFAR-10, doing classification by scaled regression.

Overall, the work is an interesting read, and a nice follow-up to Neal’s earlier observations about 1 hidden layer neural networks. It combines several insights into a nice narrative about infinite Bayesian deep networks. However, the practical utility, significance, and novelty of this work -- in its current form -- are questionable, and the related work sections, analysis, and experiments should be significantly extended.


In detail:

(1) This paper misses some obvious connections and references, such as
* Krauth et. al (2017): “Exploring the capabilities and limitations of Gaussian process models” for recursive kernels with GPs.
* Hazzan & Jakkola (2015): “Steps Toward Deep Kernel Methods from Infinite Neural Networks” for GPs corresponding to NNs with more than one hidden layer.
* The growing body of work on deep kernel learning, which “combines the inductive biases and representation learning abilities of deep neural networks with the non-parametric flexibility of Gaussian processes”. E.g.: (i) “Deep Kernel Learning” (AISTATS 2016); (ii) “Stochastic Variational Deep Kernel Learning” (NIPS 2016); (iii) “Learning Scalable Deep Kernels with Recurrent Structure” (JMLR 2017).

These works should be discussed in the text.

(2) Moreover, as the authors rightly point out, covariance functions of the form used in (4) have already been proposed. It seems the novelty here is mainly the empirical exploration (will return to this later), and numerical integration for various activation functions. That is perfectly fine -- and this work is still valuable. However, the statement “recently, kernel functions for multi-layer random neural networks have been developed, but only outside of a Bayesian framework” is incorrect. For example, Hazzan & Jakkola (2015) in “Steps Toward Deep Kernel Methods from Infinite Neural Networks” consider GP constructions with more than one hidden layer. Thus the novelty of this aspect of the paper is overstated.

See also comment [*] later on the presentation. In any case, the derivation for computing the covariance function (4) of a multi-layer network is a very simple reapplication of the procedure in Neal (1994). What is less trivial is estimating (4) for various activations, and that seems to the major methodological contribution.

Also note that multidimensional CLT here is glossed over. It’s actually really unclear whether the final limit will converge to a multidimensional Gaussian with that kernel without stronger conditions.  This derivation should be treated more thoroughly and carefully.

(3) Most importantly, in this derivation, we see that the kernels lose the interesting representations that come from depth in deep neural networks. Indeed, Neal himself says that in the multi-output settings, all the outputs become uncorrelated. Multi-layer representations are mostly interesting because each layer shares hidden basis functions. Here, the sharing is essentially meaningless, because the variance of the weights in this derivation shrinks to zero.
In Neal’s case, the method was explored for single output regression, where the fact that we lose this sharing of basis functions may not be so restrictive. However, these assumptions are very constraining for multi-output classification and also interesting multi-output regressions.

[*]: Generally, in reading the abstract and introduction, we get the impression that this work somehow allows us to use really deep and infinitely wide neural networks as Gaussian processes, and even without the pain of training these networks. “Deep neural networks without training deep networks”. This is not an accurate portrayal. The very title “Deep neural networks as Gaussian processes” is misleading, since it’s not really the deep neural networks that we know and love. In fact, you lose valuable structure when you take these limits, and what you get is very different than a standard deep neural network. In this sense, the presentation should be re-worked.

(4) Moreover, neural networks are mostly interesting because they learn the representation. To do something similar with GPs, we would need to learn the kernel. But here, essentially no kernel learning is happening. The kernel is fixed.

(5) Given the above considerations, there is great importance in understanding the practical utility of the proposed approach through a detailed empirical evaluation. In other words, how structured is this prior and does it really give us some of the interesting properties of deep neural networks, or is it mostly a cute mathematical trick?

Unfortunately, the empirical evaluation is very preliminary, and provides no reassurance that this approach will have any practical relevance:
(i) Directly performing regression on classification problems is very heuristic and unnecessary.
(ii) Given the loss of dependence between neurons in this approach, it makes sense to first explore this method on single output regression, where we will likely get the best idea of its useful properties and advantages.
(iii) The results on CIFAR10 are very poor. We don’t need to see SOTA performance to get some useful insights in comparing for example parametric vs non-parametric, but 40% more error than SOTA makes it very hard to say whether any of the observed patterns hold weight for more competitive architectural choices.

A few more minor comments:
(i) How are you training a GP exactly on 50k training points? Even storing a 50k x 50k matrix requires about 20GB of RAM. Even with the best hardware, computing the marginal likelihood dozens of times to learn hyperparameters would be near impossible. What are the runtimes?
(ii) "One benefit in using the GP is due to its Bayesian nature, so that predictions have uncertainty estimates (Equation (9)).”  The main benefit of the GP is not the uncertainty in the predictions, but the marginal likelihood which is useful for kernel learning.

---

> ### Author Response · Authors · 2017-12-20
> **Response part 2 of 2**
>
> With regards to the comments on empirical results:
>
> -- “(i) … regression on classification problems is very heuristic and unnecessary.”
>
> We do make clear that these experiments using regression for classification are less principled, in the main text. However, we’d like to note that least-squares classification is widely used and effective [3]. Moreover, it allows us to compare exact inference via a GP to prediction by a trained neural network on well-studied tasks (e.g. MNIST and CIFAR-10).
>
> -- “3) Most importantly, in this derivation, we see that the kernels lose the interesting representations that come from depth in deep neural networks. Indeed, Neal himself says that in the multi-output settings, all the outputs become uncorrelated. Multi-layer representations are mostly interesting because each layer shares hidden basis functions. Here, the sharing is essentially meaningless, because the variance of the weights in this derivation shrinks to zero.
> In Neal’s case, the method was explored for single output regression, where the fact that we lose this sharing of basis functions may not be so restrictive. However, these assumptions are very constraining for multi-output classification and also interesting multi-output regressions.”
>
> “(ii) Given the loss of dependence between neurons in this approach, it makes sense to first explore this method on single output regression, where we will likely get the best idea of its useful properties and advantages. ”
>
> This is an excellent point, which applies to almost all GP work. Based on your recommendation, we are looking into single-output regression tasks.
>
> However, we would like to emphasize that despite the NNGP being unable to explicitly capture dependencies between classes it still could outperform neural networks on multi-class regression. We believe this provides stronger, rather than weaker, evidence for the utility of the NNGP formulation.
>
> -- “(iii) The results on CIFAR10 are very poor.”
>
> First we would like to emphasize that the purpose of experiments was to show that the NNGP is the limiting behaviour for a specified neural network architecture. Because the GP equivalence was derived for vanilla fully-connected networks, all experiments were performed using that architecture. Achieving state-of-the-art on CIFAR-10 typically involves using a convolutional architecture, as well as data augmentation, batch-norm, residual connection, dropout, etc.
>
> Restricting to vanilla multi-layer fully-connected networks with ReLU activation, the performance quoted in [4] is actually slightly lower than our GP results (53-55% accuracy, Figure 4 (a) of [4]). So our baseline and results are not poor for the class of models we examine. Our experiments show that, for the given class of neural network architecture, as width increases the behaviour more closely resembles that of the NNGP, which is competitive or better than that of the given neural network class.
>
> We note that introducing linear bottleneck layer structure in [4] seem to achieve SOTA in permutation invariant (without convolutional layers) CIFAR-10 which is higher than ours. It is an interesting question how this type of model relates to the GP limit but it is outside the scope of this work.
>
> Regards to the other comments:
>
>     (i) Exact GP computation in the large data regime can be costly. We used a machine with 150 GB of RAM (with some inefficiencies in memory usage, e.g. stemming from use of float64, and TensorFlow retaining intermediate state in memory), and 64 CPU cores, to run the full MNIST/CIFAR-10 experiments. We utilized parallel linear algebra computations available through TensorFlow to speed up computations. For a typical run, constructing the kernel per layer took 90-140 seconds, and solving the linear equations (via Cholesky decomposition)  took 180-220 seconds for 1000 test points.
>
>   (ii) We agree with the reviewer that one strength of Bayesian methods is providing marginal likelihood and using that for model selection. Although we propose this possibility for future work in the text, greater emphasis could have been made. With that said, we believe that providing uncertainty estimates is another important benefit of a Bayesian approach, that we explore experimentally in our text, and that the GP perspective on neural networks is beneficial in this regard as well.
>
>
> [3] Ryan Rifkin and Aldebaro Klautau. In defense of one-vs-all classification. Journal of machine learning research, 5(Jan):101–141, 2004.
> Ryan Rifkin, Gene Yeo, Tomaso Poggio, et al. Regularized least-squares classification. Nato Science Series Sub Series III Computer and Systems Sciences, 190:131–154, 2003.
>
> [4] Zhouhan Lin, Roland Memisevic, Kishore Konda, How far can we go without convolution: Improving fully-connected networks, arXiv 1511.02580.

---

> ### Author Response · Authors · 2017-12-20
> **Response part 1 of 2**
>
> We thank the reviewer for their time and constructive feedback on the submission.
>
> -- references
>
> We thank the reviewer for suggesting related works. In the revised version, we will add Krauth et al. (2017) as well as additional comparisons with the deep kernel learning literature.
>
> -- novelty with regards to Hazan & Jaakola (2015)
>
> We do not believe that the work in H-J significantly detracts from the novelty of our paper.
>
> H-J is also interested in constructing kernels equivalent to infinitely wide deep neural networks. Theorem 1 in H-J is a good stepping stone for our construction. However the H-J construction does not go beyond two hidden layers with nonlinearities. They state:
> “We present our framework with only two intermediate layers ... It can be extended to any depth but the higher layers may not use nonlinearities." H-J
>
> We believe that the fact that H-J approached the same problem, and only derived a GP kernel for up to two layers, despite making use of the same random kernel literature we do, is illustrative of the non-obvious nature of the equivalence between infinitely wide networks of arbitrary depth and GPs.
>
> We will expand our existing discussion of H-J in the text, and state that previous work has proposed GP kernels for networks with up to two hidden layers.
>
> --“In any case, the derivation for computing the covariance function (4) of a multi-layer network is a very simple reapplication of the procedure in Neal (1994).”
>
> We agree that the derivation is simple. We believe that this, combined with the fact that it has gone unpublished for more than two decades, increases rather than detracts from its significance.
>
>  --“Also note that multidimensional CLT here is glossed over. It’s actually really unclear whether the final limit will converge to a multidimensional Gaussian with that kernel without stronger conditions. This derivation should be treated more thoroughly and carefully.”
>
> Thank you for sharing your concerns. We would very much like to address them. Could you be more specific about the ways in which you are concerned the CLT may fail in this case? If we take the infinite-width limit layer-by-layer, the application of the CLT seems appropriate without additional subtlety.
>
> -- “In fact, you lose valuable structure when you take these limits, and what you get is very different than a standard deep neural network. In this sense, the presentation should be re-worked.”
>
> We agree that the qualitative behavior of infinitely wide neural networks may be different than that of narrow networks. We will update the text to more clearly discuss this.
>
> We note though that finite width network performance often increases with increasing network width, as they become closer to the GP limit. For example, see [1], [2]. In fact in our Figure 1, we found that the performance of finite width networks increases, and more closely resembles that of the NNGP, as the network is made wider.
>
> To more thoroughly address this concern and support this observation, we performed an additional experiment where we trained 5 layer fully connected networks with Tanh and ReLU nonlinearities on CIFAR10, with random optimization and initialization hyperparameters. We then filtered for training runs which achieved 100% classification accuracy on the training set, resulting in 125 Tanh and 55 ReLU networks. We then examined the performance of these networks vs. network width. We found that the best performing networks are in fact the widest. See the following figures, where each point shows the width and corresponding generalization gap of a single trained 5 layer network, with 100% training accuracy:
> https://www.dropbox.com/s/np4myfzy1a3ts46/relu_depth_5_gap_to_width_cifar10.pdf
> https://www.dropbox.com/s/f1cd73hvpesm8n2/tanh_depth_5_gap_to_width_cifar10.pdf
>
> -- “Moreover, neural networks are mostly interesting because they learn the representation. To do something similar with GPs, we would need to learn the kernel. But here, essentially no kernel learning is happening. The kernel is fixed.”
>
> We agree that the learned representations are one important aspect of deep networks, and we agree that no explicit representation learning happens in our GP approach.
>
> However, we emphasize that in many situations deep networks are chosen not for their interpretable representations, but rather because of the high accuracy of their predictions. We believe that work that reproduces the predictions made by deep networks using an alternative procedure is useful even if it does not also reproduce the internal representations of deep networks.
>
> Ways to sample deep representations from the corresponding NNGP would be a fascinating avenue for future research.
>
> [1] Neyshabur B, Tomioka R, Srebro N. In search of the real inductive bias: On the role of implicit regularization in deep learning. arXiv:1412.6614. 2014.
>
> [2] Zagoruyko S, Komodakis N. Wide residual networks. arXiv:1605.07146. 2016.

---

### Official Review · AnonReviewer2 · 2017-11-28
**Interesting characterization of deep Bayesian NNs, but not the most interesting regime for deep learning.**

**Rating:** 6
**Confidence:** 4

**Review:**

This paper leverages how deep Bayesian NNs, in the limit of infinite width, are Gaussian processes (GPs). After characterizing the kernel function, this allows us to use the GP framework for prediction, model selection, uncertainty estimation, etc.


- Pros of this work

The paper provides a specific method to efficiently compute the covariance matrix of the equivalent GP and shows experimentally on CIFAR and MNIST the benefits of using the this GP as opposed to a finite-width non-Bayesian NN.

The provided phase analysis and its relation to the depth of the network is also very interesting.

Both are useful contributions as long as deep wide Bayesian NNs are concerned. A different question is whether that regime is actually useful.


- Cons of this work

Although this work introduces a new GP covariance function inspired by deep wide NNs, I am unconvinced of the usefulness of this regime for the cases in which deep learning is useful.

For instance, looking at the experiments, we can see that on MNIST-50k (the one with most data, and therefore, the one that best informs about the "true" underlying NN structure) the inferred depth is 1 for the GP and 2 for the NN, i.e., not deep. Similarly for CIFAR, where only up to depth 3 is used. None of these results beat state-of-the-art deep NNs.

Also, the results about the phase structure show how increased depth makes the parameter regime in which these networks work more and more constrained.

In [1], it is argued that kernel machines with fixed kernels do not learn a hierarchical representation. And such representation is generally regarded as essential for the success of deep learning.

My impression is that the present line of work will not be relevant for deep learning and will not beat state-of-the-art results because of the lack of a structured prior. In that sense, to me this work is more of a negative result informing that to be successful, deep Bayesian NNs should not be wide and should have more structure to avoid reaching the GP regime.


- Other comments:

In Fig. 5, use a consistent naming for the axes (bias and variances).

In Fig. 1, I didn't find the meaning of the acronym NN with no specified width.

Does the unit norm normalization used to construct the covariance disallow ARD input selection?


[1] Yoshua Bengio, Olivier Delalleau, and Nicolas Le Roux. The Curse of Dimensionality for Local Kernel Machines. 2005.

---

> ### Author Response · Authors · 2017-12-20
> **Response part 2 of 2**
>
> -- Fixed Kernel machines vs representation learning of deep neural networks
>
> While the functional form of our GP kernel is fixed, and no kernel learning is happening in the sense of Deep Kernel Learning [3], we do learn hyper-parameters (induced by neural network architecture) for kernels in our experiments by grid search. Using GP marginal likelihood, one could learn hyper-parameters for the equivalent neural network by end-to-end gradient descent as well.
>
> Although our NNGP does not admit explicit hierarchical representation learning, we note that our experiments showing that an NNGP can perform better than its finite width counterpart suggest interesting scientific question on the role of learned representations. Exploring ways to sample intermediate representations from the posterior implied by the NNGP would be a fascinating direction for future work.
>
> Regards to the other comments:
>
> -- In Fig. 5, use a consistent naming for the axes (bias and variances).
>
> Thank you for noticing this. We will update the figures in the revised version.
>
> -- In Fig. 1, I didn't find the meaning of the acronym NN with no specified width.
>
> We will include the description in the revised version. The acronym NN in the figure denotes the best performing (on the validation set) neural network across all width and trials. Often this is the neural network with the largest width.
>
> -- “Does the unit norm normalization used to construct the covariance disallow ARD input selection?”
>
> Thank you for bringing up the point about ARD. With extra computational and memory cost, unit normalization for inputs could be avoided by separately tiling the variance of each input when constructing the lookup table in Section 2.5. Also note, input pre-processing in general can change ARD scores, and scaling inputs to have a constant norm is not an uncommon form of pre-processing.
>
>
> Thank you again for your careful review! We believe we have effectively addressed your primary concern about the relevance of the wide network limit, and we hope you will consider raising your score as a result.
>
> [3] Andrew Gordon Wilson, Zhiting Hu, Ruslan Salakhutdinov, Eric P. Xing, Deep Kernel Learning. AISTATS 2016.

---

> ### Author Response · Authors · 2017-12-20
> **Response part 1 of 2**
>
> We thank the reviewer for their time and constructive feedback on the submission.
>
> -- Usefulness of the regime.
>
> As noted, the best performing depth for the NNGP for full datasets was shallow (depth 1 in MNIST and depth 3 in CIFAR-10). A few points about this:
>   i) For these datasets, the best performing neural network is also shallow (depth 2 in MNIST and depth 3 or 2 in CIFAR-10). As our NNGP construction is the limiting behaviour of wide neural networks, we believe that the GP performing best with a shallow depth is consistent with this equivalence.
>   ii) For the small data-regime, we note that there were benefits from increased depth, both for the NN and the NNGP.
>   iii) We also note that when the dataset became more complex (MNIST to CIFAR-10) the GP and NN both benefited from additional depth.
>   iv) All experiments in our paper were performed in the fully connected case, where the evidence for the benefits of hierarchy+depth is weaker than for convolutional networks.
>   v) Lastly, although the best performing depth are shallow, the deep NNGPs perform competitively with the shallow ones. For example, with RELU the depth 10 NNGP for MNIST-50k has test accuracy of 0.987, and for CIFAR-45k with RELU has test accuracy 0.5573. The best performing accuracy for those cases was 0.9875 and 0.5566 respectively. (Note that for CIFAR depth 10 test accuracy is actually *higher* than depth-3, this is due to model selection based on the validation set rather than the test set.) The performance loss from depth in NNs is much larger, possibly due to harder optimization.
>
> -- “In [Bengio, Delalleau, and Le Roux], it is argued that kernel machines with fixed kernels do not learn a hierarchical representation. And such representation is generally regarded as essential for the success of deep learning.
>
> My impression is that the present line of work will not be relevant for deep learning and will not beat state-of-the-art results because of the lack of a structured prior. In that sense, to me this work is more of a negative result informing that to be successful, deep Bayesian NNs should not be wide and should have more structure to avoid reaching the GP regime.”
>
> First, we would like to note that finite width network performance often increases monotonically with increasing network width, as the networks become closer to a GP limit. For example, see [1], [2]. In fact in our Figure 1, we found that the performance of finite width networks increases, and more closely resembles that of the NNGP, as the network is made wider.
>
> To more thoroughly address this concern and support this observation, we performed an additional experiment where we trained 5 layer fully connected networks with tanh and ReLU nonlinearities on CIFAR10, with random optimization and initialization hyperparameters. We then filtered for training runs which achieved 100% classification accuracy on the training set, resulting in 125 Tanh and 55 ReLU networks. We then examined the performance of these networks vs. network width. We found that the best performing networks are in fact the widest. See the following figures, where each point shows the width and corresponding generalization gap of a single trained 5 layer network, with 100% training accuracy:
> https://www.dropbox.com/s/np4myfzy1a3ts46/relu_depth_5_gap_to_width_cifar10.pdf
> https://www.dropbox.com/s/f1cd73hvpesm8n2/tanh_depth_5_gap_to_width_cifar10.pdf
>
> Second, we would like to address the concerns about kernel methods which the reviewer cites from Bengio, Delalleau, and Le Roux (BDL). The analysis in BDL assumes a local kernel (e.g. an RBF kernel). The NNGP kernel is non-local and heavy tailed, as can be seen in the Figure showing its angular structure in Appendix B of our paper. Specifically, Equation 10 in BDL, which demands that the kernel approaches a constant with increasing distance between points, does not hold for the NNGP kernel: As discussed in our paper, inputs are scaled to have constant norm -- i.e. all inputs live on the surface of a hypersphere. There is no angular separation between points on the hypersphere after which the NNGP kernel goes to a constant (again see Appendix B figure).
>
> Finally, we are not sure if we have fully understood your concern about the lack of a structured prior. If the above responses do not address your concern, could you be more specific about what structure is required in a prior of functions which a GP is unable to capture?
>
> [1] Neyshabur B, Tomioka R, Srebro N. In search of the real inductive bias: On the role of implicit regularization in deep learning. arXiv:1412.6614. 2014.
>
> [2] Zagoruyko S, Komodakis N. Wide residual networks. arXiv:1605.07146. 2016.

---

### Official Review · AnonReviewer3 · 2017-11-29
**Well written and insightful**

**Rating:** 7
**Confidence:** 3

**Review:**

This paper presents a new covariance function for Gaussian processes (GPs) that is equivalent to a Bayesian deep neural network with a Gaussian prior on the weights and an infinite width. As a result, exact Bayesian inference with a deep neural network can be solved with the standard GP machinery.


Pros:

The result highlights an interesting relationship between deep nets and Gaussian processes. (Although I am unsure about how much of the kernel design had already appeared outside of the GP literature.)

The paper is clear and very well written.

The analysis of the phases in the hyperparameter space is interesting and insightful. On the other hand, one of the great assets of GPs is the powerful way to tune their hyperparameters via maximisation of the marginal likelihood but the authors have left this for future work!


Cons:

Although the computational complexity of computing the covariance matrix is given, no actual computational times are reported in the article.

I suggest using the same axis limits for all subplots in Figure 3.

---

> ### Author Response · Authors · 2017-12-20
> **Response**
>
> Thank you for your time and constructive suggestions on the submission.
>
> -- “Although the computational complexity of computing the covariance matrix is given, no actual computational times are reported in the article.”
>
> We are grateful the suggestion. In the revised version we will add computation time for full MNIST for reference. As one datapoint, when constructing the 50k x 50k covariance matrix, the amortized computations for each layer take 90 -140s (depending on CPU generation and network depth), running on 64 CPUs.
>
> --”I suggest using the same axis limits for all subplots in Figure 3.”
>
> We will update the figures accordingly in the revised version.

---

### Public Comment · (anonymous) · 2017-12-15
**Reproducibility Assessment**

Availability of Requisite Information: The appendix of this paper does a good job outlining most of the necessary information for reproducing the results. The computation of the Gaussian process is thorough and well described, with the exception of some information regarding the lookup table. The values n_g, n_v, and n_c were all provided, however a value for u_max was not provided, nor was it explained how it was determined. With regards to testing, the authors provide the neural network and Gaussian process configurations used to obtain their best results, as well as their random search ranges for various hyper-parameters. Furthermore, the data sets on which they trained and tested their algorithms are publicly available and readily accessible.

Computational Resources Required: The Gaussian process algorithms were run on a laptop with 8 GB of RAM, and a dual core 2.9GHz processor. The kernel matrix for the full MNIST training set is over 20 GB. Furthermore, this matrix needs to be inverted for prediction. Due to memory requirements, the results for 20,000 and 50,000 training points could not be computed. With matrix multiplication and parallel computing, the lookup table, which theoretically is O((n_g)^2 n_v n_c), took just under an hour to calculate for the values n_g = 500, n_v = 501,and n_c = 500.The accuracies for the Gaussian processes in Table 2 were all computed (with the exception of 20,000, and 50,000 sizes) overnight. Finally we estimated that reproducing Figure 7, MNIST=5,000, d=50, tanh would have taken us approximately 60 hours.

Prior Knowledge Required: The paper does a good job giving an overview of the theoretical results at the beginning of the paper. They outline how to construct the kernel, as well as how to use the kernel to do prediction. However, the value for u_max perhaps required a deeper theoretical understanding to determine. Further, the lookup table algorithm is not well-defined. For values of c = ±1and s = 0,the lookup table is defined with the inverse of a singular matrix. The case of c = 1is addressed, however the solution to these singularities is not explicitly given.

Results: The Gaussian process calculations were reproducible, although the accuracies we obtained did not perfectly match those provided in the paper. Notably, the LUT only gave good accuracy on MNIST with training data size larger than 2000. The replicated baseline accuracies overall were quite close to those stated in the paper. They typically were slightly lower, likely due to the differences in hyperparameter optimization. The authors of the paper had access to Google’s Vizier technology, which is not universally available. Despite this, using the same model depths and widths, values within ~5% were obtainable.

Conclusions: The time for reproducing Table 2 was reasonable, though we would not have been able to reproduce the optimization of the GP hyper-parameters given our computational resources. We were also unable to reproduce the deep-signal propagation heat maps given our resources and time constraints. There is nothing to suggest these results would not have been reproducible, however, given additional resources. The numerical kernel implementation is arguably the central contribution of this paper. Given the information provided in the paper, we could not reproduce the results from this numerical algorithm. It would have helped to have the original source code to guide us, especially with respect to handling the singular cases of the lookup table. The source code does not seem to be available online. We have contacted the authors to ask whether it was or could be made available, but we had not received an answer by the time of writing. In conclusion, this paper was fairly reproducible, however requires a high level of computational power and theoretical knowledge. Specifying u_max and numerically stable solutions to their lookup table would have aided in the paper’s reproducibility.

Our full report can be found at: https://github.com/niklasbrake/COMP-551-Project-4/blob/master/Final%20Report.pdf

---

> ### Author Response · Authors · 2018-01-05
> **Response**
>
> (We have been in email correspondence with the commenters of the reproducibility assessment; the following details our correspondence, provided for clarification.)
>
> Thank you for the interest in our work and putting careful effort to reproduce our results! To clarify, we haven’t made our research code public yet, but, as denoted in the submission, are working towards making it open sourced.
>
> To address some specific concerns in the report:
>
> -- “Notably, the LUT only gave good accuracy on MNIST with training data size larger than 2000.”
>
> We are surprised to hear that. Looking into the github codebase, we’ve noticed two possible causes.
>
> (1). First, we’ve noticed that GP regression was done using the actual inverse of K_DD matrix. In practice taking the inverse for solving linear systems equations is known to be numerically unstable, especially when the condition number is large. Especially in Table 2, the smaller dataset’s best performing depth was quite high (100) and deeper kernels become more degenerate requiring careful numerics. Results of pure random accuracy (0.1) is one outcome of unstable numerics. In practice, we used Cholesky decomposition to solve linear systems equations, which is faster and numerically more stable. Also relatedly adding \sigma_epsilon to K_DD helps with numerical stability. We thank you for noticing that we do not report the value we use in our experiments. We have been using 1e-10 as our default value and kept multiplying it by 10 when Cholesky decomposition failed due to numerical instability.
>
> In general, we recommend using Cholesky decomposition utilizing positive semi-definiteness of the covariance matrix, or linear systems solver / pseudo-inverse to make regression stable.
>
> (2). Second, it appears that numerical values for \sigma_w and \sigma_b used in the code seem to be \sigma_w^2, \sigma_b^2 (variance instead of std. deviation) in our paper. If this is the case, the poor performance for deeper kernels is understandable. The phase diagram for deep networks show that the (\sigma_w^2, \sigma_b^2) pair is quite sensitive in obtaining good performance. Numerical values for variance to standard deviation will be quite different, and we are worried that this might have caused not obtaining as good a performance as ours.
>
> We thank you for pointing out information that wasn’t readily provided in the paper. We’ll incorporate your suggestions to make the paper more easily reproducible.
>
> In regards to u_max, the value you’ve chosen (18) should to be good. In our experiments we’ve used 10 but after submission to ensure numerical accuracy for all range of variance (0, s_max =100), we preferred using larger u_max of either 50 or 100.
>
> Also, regarding c_j = 1, in the lookup table we restrict ourselves to | c_j |<0.99999. The footnote part refers to the diagonal elements (variance), where we separately performed 1d Gaussian integral over \phi(z)^2 to construct one dimensional grid and interpolate with it.
>
> Thank you again for your careful assessment! We are grateful that you chose our paper and made valuable suggestions to make our paper easier to reproduce. Also, we encourage you to  see if the numerically stable linear algebra solver and correcting values for (\simga_w^2, \sigma_b^2) would bring your GP results close to what we obtain and more competitive to the neural network results.

---

### Author Response · Authors · 2018-01-05
**Summary of Paper Updates**

Below is a summary of the most salient revisions:

—updated discussion of relevant work suggested by reviewers
—more discussion of implementation details (e.g. computation time)
—figures with updated axes ranges/labels or captions
—citation to a parallel ICLR 2018 submission, “Gaussian Process Behavior in Wide Deep Neural Networks.”
—an added appendix describing an alternative derivation of the GP correspondence via marginalization over intermediate layers

---

### Decision · Program_Chairs · 2018-01-29
**ICLR 2018 Conference Acceptance Decision**

**Decision:**

Accept (Poster)

**Comment:**

This paper presents several theoretical results linking deep, wide neural networks to GPs.  It even includes illuminating experiments.

Many of the results were already developed in earlier works. However, many at ICLR may be unaware of these links, and we hope this paper will contribute to the discussion.